# Acute Effects of Remedial Exercises with and without Compression on Breast-Cancer-Related Lymphedema

**DOI:** 10.3390/healthcare11222949

**Published:** 2023-11-11

**Authors:** Gülbala Gülören, Yahya Doğan, Serap Özgül, Ceren Gürşen, Gamze Nalan Çinar, Funda İpekten, Türkan Akbayrak

**Affiliations:** 1Faculty of Physical Therapy and Rehabilitation, Hacettepe University, Ankara 06100, Turkey; yahyadogan111@hotmail.com (Y.D.); serapky@yahoo.com (S.Ö.); cerengursen@yahoo.com (C.G.); nalan.gd@gmail.com (G.N.Ç.); takbayrak@yahoo.com (T.A.); 2Department of Biostatistics, Faculty of Medicine, Erciyes University, Kayseri 38039, Turkey; fundaipekten@gmail.com

**Keywords:** breast cancer, lymphedema, exercise, bioimpedance spectroscopy

## Abstract

Remedial exercises are an important part of the treatment for lymphedema, but there is little evidence to support the acute effects of remedial exercises with or without compression. The aim of this study was to investigate whether and how daily (performed within 24 h) remedial exercises with and without compression bandaging acutely affect the severity of lymphedema and its symptoms in breast-cancer-related lymphedema (BCRL). In total, 34 patients with BCRL completed three sets of remedial exercises (within 24 h) with and without compression bandaging in a randomized order separated by a 3-day wash-out period. The severity of lymphedema and extracellular water ratio were assessed before and 24 h post exercise by using bilateral circumferential measurements and bioimpedance spectroscopy (in L-dex score), respectively, and the severity of self-reported symptoms (swelling, heaviness, and tightness) was assessed using a visual analogue scale. While there was no difference in all outcomes at 24 h post exercise without compression (*p* > 0.05), all outcomes decreased significantly compared to baseline at 24 h after the exercise with compression (*p* < 0.001). The remedial exercises performed in the absence of compression within 24 h do not acutely increase the lymphedema and symptoms in BCRL. These are important preliminary findings, which can be used to inform future prospective evaluation of the long-term effects of remedial exercise performed without compression.

## 1. Introduction

Breast-cancer-related lymphedema (BCRL) is one of the adverse effects of breast cancer treatment [1]. Lymphedema is a progressive and persistent chronic inflammatory condition caused by damage or overload to the lymphatic system that negatively affects patients’ work, career, health-related quality of life (HRQoL), daily activities, and functionality [2,3]. Although lymphedema can occur in the breast, upper and lower arm, chest, and/or trunk of the affected side due to the mechanically insufficient lymphatic transport capacity, it most commonly appears in the upper extremity of the affected side [3]. BCRL may occur due to trauma or infections after cancer treatments including axillary lymph node dissection or radiotherapy [4]. An incidence of 14–54% has been reported for secondary upper-extremity lymphedema after breast cancer treatment in patients who have undergone breast cancer surgery (i.e., mastectomy and axillar lymph node dissection), radiotherapy, and chemotherapy [5]. However, there is a consensus that the prevalence of BCRL changes depending on more extensive adjuvant treatments, more invasive and radical surgical techniques, and patient-related risk factors (i.e., age, increased body mass index, low physical activity level, and sedentary lifestyle) [6,7]. According to a previous study integrating machine learning technology into determining risk factors in breast cancer survivors, the number of metastatic lymph nodes, breast cancer grading, presence of lymphovascular invasion, type of estrogen therapy before breast cancer, age of the patient at diagnosis, trastuzumab therapy, human epidermal growth factor receptor 2 (HER2), and taxane-based chemotherapy have been characterized as the most important risk factors for BCRL development [8]. Lymphedema can cause a decrease in muscle strength and range of motion of the shoulder, sensory disturbances, hypersensitivity, fatigue, and physical symptoms such as swelling, pain, heaviness, tightness, and discomfort, which can lead to functional upper-limb impairments [3,6]. BCRL also causes psychological symptoms such as a negative body image, low self-esteem, poorer social well-being, and emotional problems such as stress, anxiety, and kinesiophobia associated with the increased severity of the lymphedema [3,9,10].

Since lymphedema is a complex condition, multidisciplinary approaches combining conservative and surgical techniques are required in the treatment of BCRL. In the surgical management of BCRL, physiological techniques (lymphatic–venous anastomosis and vascularized lymph node transfer) are more effective in mild/moderate lymphedema, while ablative techniques are more necessary in severe lymphedema characterized by trophic skin changes such as fibrosis, thickness, and warty overgrowths [11]. In order to preserve postoperative results, it is recommended to combine pre- and postoperative conservative modalities with both physiological and ablative surgical techniques [11,12].

Currently, several conservative interventions exist for lymphedema management in the literature; however, there is limited consensus on the optimal conservative modality of BCRL [12]. Among the conservative approaches, complex decongestive physiotherapy (CDP), consisting of patient education, manual lymphatic drainage, compression therapy, remedial exercises, and skin care, is the most recommended method to improve HRQoL [3,11]. Additionally, exercise interventions such as supervised aerobic exercise and progressive resistance exercise have been reported to have significant potential effects on the prevention of BCRL [13].

The American Society of Clinical Oncology reported that cancer survivors could safely engage in exercise training to improve cancer-related health issues, including anxiety, depressive symptoms, cancer-related fatigue, physical functioning, body composition, and HRQoL [14]. A growing body of literature provides evidence that these improvements also occur in patients with BCRL [15]. Remedial exercises, an integral component in CDP, are specific types of physical activities designed to repeatedly compress the lymph vessels through rhythmic and repetitive muscle contraction and relaxation in the involved area [16]. Remedial exercises with compression garments or bandages also enhance the muscle pump for increased venous and lymphatic fluid return to the circulatory system and out of the swollen areas [17]. The clinical guidelines recommend that exercise should be performed with compression in order not to increase lymphedema; however, the evidence behind this clinical recommendation is unclear [16]. In addition, patients with lymphedema report problems with constant compliance with compression therapy, another component of CDP, for social and physical reasons and request short-term compression treatments or short breaks from ongoing compression therapy. However, to the best of our knowledge, there is no evidence of whether remedial exercises alone acutely exacerbate lymphedema in at-risk individuals or of how the acute effects of remedial exercises with compression maintain reduction in the severity of lymphedema and improvement in self-reported lymphedema symptoms. Therefore, we aimed to assess the acute effects of remedial exercises, with and without compression, on the severity and symptoms of lymphedema in BCRL through limb volume/circumference and extracellular measurements and also via self-reported outcomes.

## 2. Materials and Methods

### 2.1. Participants

This study utilized a prospective, randomized, and cross-over design. The study protocol was approved by the Hacettepe University Clinical Research Ethics Boards (Approval number: KA-22053) and was registered at ClinicalTrials.gov (NCT05610579).

The inclusion criteria were as follows: (1) being older than 18 years of age, (2) having a diagnosis of BCRL through objective measurements (i.e., circumferential measurement and bioimpedance spectroscopy), (3) having >2 cm differences between the extremities according to the circumferential measurement or having an L-Dex score ≥ 7, (4) having breast cancer treatments (including surgery, radiotherapy, and chemotherapy) at least 3 months before participation in the study, (5) volunteering to participate in the study, (6) having stage II to III BCRL according to the International Society of Lymphology, and (7) scheduled for complex decongestive therapy but have not yet started complex decongestive therapy. Exclusion criteria were as follows: (1) having tumor metastasis, (2) having symptoms or signs of non-breast-cancer-related (lymph)edema, (3) having uncontrolled diabetes mellitus, (4) having pre-existing neuromusculoskeletal or neurological conditions of the upper extremities, (5) having active infection in the affected extremity (fever, redness, increased skin temperature or swelling, and pain), (6) having a bilateral breast surgery, (7) experiencing contraindications for compression bandage (including cardiac edema, peripheral neuropathy, systemic edema, and malignity), and (8) not being able to communicate.

### 2.2. Assessment and Outcome Measures

After baseline assessment, each participant completed the remedial exercise with a compression bandage and without compression bandaging in a randomized order separated by a 3-day wash-out period. Participants were randomized to determine which exercise intervention was initiated first in an allocation ratio of 1:1 using a computer-based block randomization procedure (Sealed Envelope Ltd., 27–31 Clerkenwell Close, London, UK) carried out by C.G., who had no role in data collection.

Demographic, physical, and medical characteristics of the patients were recorded. As outcome assessments, extremity circumference, the ratio of extracellular water, and self-reported lymphedema symptoms were assessed/asked about before and 24 h after each exercise session by the same physiotherapist. The reference points for circumferential and bioimpedance spectroscopy (BIS) assessments were marked with a waterproof tattoo pen.

#### 2.2.1. Arm Volume Based on Circumferential Measurements

The circumferential measurements were taken bilaterally with a tape from the ulnar styloid process to the axilla, with 5 cm intervals. Volume was estimated from circumferential values using the truncated cone formula, which yielded excellent inter- and intra-observer reproducibility (0.97 and 0.98, respectively) in comparison to water displacement [18]. To calculate the relative excess arm volume, the following formula was used: [(volume of the edematous limb—volume of the non-edematous limb)/(volume of the non-edematous limb)] × 100 [19,20]. According to the literature, the non-dominant arm is on average 3.3% smaller than the dominant arm. To correct for arm dominance, the arm volume of the non-dominant arm was adjusted by 3.3% [20]. In addition, after correction of the arm dominance, the severity of lymphedema was determined using the difference between the estimated volumes of the affected and unaffected arms (mild lymphedema: 200–250 mL volumetric difference; moderate lymphedema: 250–500 mL difference; severe lymphedema: >500 mL difference) [21].

#### 2.2.2. Bioimpedance Spectroscopy Measurement

The extracellular water ratio was used to detect tissue changes as a consequence of the severity of lymphedema, which was assessed using bioimpedance spectroscopy (BIS) (Impedimed, L-Dex U400, Pinkenba, Queensland, Australia) [22]. BIS is also used to determine changes specific to the volume of extracellular fluid that contains lymph. Participants were asked to adopt a supine position with their arms slightly abducted. The dominant limb and affected limb were recorded on a device, and metal accessories on the extremities were removed. Before performing the measurement, the skin under the electrode placed on the hand, arm, and foot to be measured was cleaned with an alcohol wipe [23,24]. Participants were classified as having lymphedema if their inter-limb BIS ratio exceeded at least one of these previously reported thresholds based on normative population: exceeding ≥ 7SD (standard deviation) above the mean for the whole arm [25]. The measurements were performed on the right-hand side, followed by the left-hand side, to measure the impedance of the extracellular fluid for each limb, with the ratio of these values comparing the affected and non-affected limbs. Finally, these measurements were later converted to another metric, the L-Dex value.

#### 2.2.3. Severity of Self-Reported Lymphedema Symptoms

The severity of the most commonly self-reported lymphedema symptoms, including swelling, heaviness, and tightness, was assessed [26]. Participants marked on a 10 cm visual analogue scale (VAS) the extent to which they perceived their arm as swollen, heavy, or tight during the past month, with 0 being “not at all” and 10 being “extremely” swollen/puffy, heavy, or tight. The distances of the marked points to the 0 point were recorded in cm [27]. The VAS has been shown to be a valid and reliable tool for measuring self-reported symptoms, and it is also sensitive to changes with treatment of lymphedema and to small changes in symptom intensity [28].

### 2.3. Interventions

The two interventions comprising remedial exercises without compression bandaging and remedial exercises with compression bandaging were performed by all participants in a randomized order separated by a 3-day wash-out period. After the baseline/first assessment, one set of remedial exercises with and without compression was performed under the supervision of the physiotherapist. The participants were asked to complete an additional three sets of remedial exercises at home within the following 24 h. At the same time on the next day, in the 24th hour, the patients were evaluated a second time (if there was a compression bandage, after it was removed). In the following wash-out period, all participants performed their normal lymphedema management/care strategies. After 3 days, following the 3rd evaluation in the clinic, remedial exercises with and without compression were performed again, and 24 h later, the 4th evaluation was performed in the same way. Additionally, participants were asked to maintain their dietary habits and physical activity levels throughout the study period.

#### 2.3.1. Remedial Exercise Program

The following exercises (10 repetitions of each exercise in one set) were given to patients by a qualified physiotherapist after practicing together: (1) deep abdominal breathing; (2) warm up activity of active mobilization of upper extremity joints at a moderate pace (shoulder girdle mobilization, flexion, extension and circumduction of shoulder, elbow and wrist joints, and ball squeeze); (3) distal pumping exercise in elevation (using a metaphor, i.e., “as if taking something from the high closet shelf” and “as if picking an apple from the tree”); (4) the Diagonal 1 (D1) flexion and extension pattern of the progressive neuromuscular exercise (PNF) (using a metaphor, i.e., “as if throwing a scarf across the opposite shoulder”); (5) the Diagonal 2 (D2) flexion and extension pattern of the PNF (using “take out a sword” metaphor); (6) stretching the pectoralis and trapezius muscles [26,29,30]. All exercises were performed in an upright position.

Participants received a booklet about the exercise program after initial education, and a home exercise diary was given to participants for facilitating adherence. The exercises were also checked by the physiotherapist through the online sessions during the day.

#### 2.3.2. Compression Therapy

Multilayered compression bandaging sessions were performed by the same physiotherapist. Compression was provided with stockinettes (Tricofix^®^; BSN Medical, Luxembourg), padded foam bandages (Artiflex^®^; BSN Medical), and short elastic bandage materials (Comprilan^®^; BSN Medical).

### 2.4. Statistical Analysis

Statistical analyses were performed by using the Statistical Package for the Social Sciences software, version 25.0 (IBM SPSS Statistics; IBM Corporation, Armonk, NY, USA). Normal distribution of the variables was tested using histograms, Q–Q plots, and Shapiro–Wilk tests. For descriptive statistics, mean ± standard deviation, median (IQR: 25th–75th percentiles), and minimum–maximum values were used for numerical variables, while numbers and percentages were used for categorical variables. The Pearson correlation analysis was used to determine the relationship between the numerical variables. The paired-samples *t* test was used for numerical variables in repeated pairwise comparisons. *p* < 0.05 was considered statistically significant.

### 2.5. Sample Size Calculation

Sample size was estimated using the G*Power (Version 3.1.9 for Mac) program. In a study [31] that used low-intensity resistance exercise and compression with socks in patients with BCRL and compared “baseline”, “immediately post-exercise”, and “24 h post-exercise” values with and without compression therapy, a strong effect size [32] (d = 0.57) was obtained for lymphedema symptoms. Based on that study, it was calculated that at least 32 individuals should be included in the present study, with a confidence level of 95%, a power of 80%, and a drop-out rate of 20%.

## 3. Results

Between November 2022 and February 2023, 45 participants in total were screened for eligibility. Of these, 8 participants did not meet the inclusion criteria (insufficient literacy (n = 1), infection signs (n = 2), bilateral mastectomy (n = 2), diabetes mellitus (n = 2), and malignity recurrence (n = 1)). Three participants were excluded from the study, as they did not participate in the second intervention period. Consequently, thirty-four women with BCRL (mean age 59.8 ± 8.78 years; mean body mass index 29.6 ± 4.21 kg/m^2^; mean lymphedema duration 47.5 ± 34.9 months) met the inclusion criteria and completed the study with full exercise compliance. Figure 1 shows the study flow-chart. Sixteen participants were randomized to begin remedial exercises without a compression bandage first, and eighteen were randomized to begin with a compression bandage. Descriptive characteristics of the participants are given in Table 1.

### 3.1. Severity of Lymphedema

Pre- to 24 h post-exercise changes in limb volume and BIS ratio are shown in Table 2. There were no significant differences in the affected total arm volume (mean (SD) arm volume: 2977.9 ± 533.8 cm^3^) or between the affected and unaffected side volume (cm^3^) and relative arm volume (%) from before to 24 h after the remedial exercise without compression (*p* > 0.05). Table 2 also shows that the impedance ratio (L-Dex score) remained the same from before to 24 h after the remedial exercise without compression (*p* > 0.05). However, there was a significant decrease in the L-Dex score 24 h after the remedial exercise with compression bandage (*p* < 0.001). Additionally, the change in the affected total arm volume and the difference between the affected and unaffected side volume (cm^3^) and relative arm volume (%) decreased significantly compared with the baseline and after the remedial exercise with compression bandage (*p* < 0.001).

### 3.2. Severity of Symptoms

Changes in self-reported lymphedema symptoms from before to 24 h after the remedial exercise with and without compression are provided in Table 3. The change in VAS mean scores of all symptoms (swelling, heaviness, and tension) decreased significantly 24 h after the remedial exercise with compression bandage (*p* < 0.001). No significant changes were found from before exercise to 24 h post remedial exercise without compression in the symptoms of swelling, heaviness, and tightness according to the VAS mean scores (*p* > 0.05).

### 3.3. Comparison Results of Percentage Changes of Remedial Exercises with and without Compression

When the effects of exercise with and without compression were compared between groups, there were statistically significant differences in the changes in all outcome measures (*p* < 0.05). It was found that the amount of percentage change in all symptoms (swelling, heaviness, and tightness), the limb volume measurements, and extracellular water ratio variables in the remedial exercise with compression group were greater than the amount of percentage change in the remedial exercise without compression group (Table 4).

## 4. Discussion

According to the results of this study, the severity of lymphedema, the relative lymphedema ratio (in %), and the extracellular water ratio and severity of self-reported lymphedema symptoms did not increase within 24 h post exercise when remedial exercises were performed without a compression bandage. Additionally, it is demonstrated that daily (performed within 24 h) remedial exercises with compression bandaging provided a significant reduction in the severity of lymphedema and lymphedema-related symptoms. The present study highlights the need to better understand issues such as compression therapy (bandage or sleeve) surrounding the knowledge of remedial exercise as considered to be a key aspect of lymphedema treatment and recognition of the importance of daily home-based remedial exercise and the impact of adherence on study outcomes. To the best of our knowledge, the present study is the first to investigate the 24 h acute effects of remedial exercises in women with BCRL. Additionally, we aimed to investigate whether remedial exercises with compression bandaging affect the severity of lymphedema and self-reported lymphedema symptoms in 24 h.

Until the early 2000s, the use of compression bandages or garments during waking hours and physical activities was the essential issue in the management of BCRL [33]. Additionally, women with or at risk of BCRL were advised to avoid strenuous upper-extremity activity and/or resistance exercise, especially including the affected upper extremity because of the risk that such activity could lead to the exacerbation of lymphedema [34]. In the traditional treatment of lymphedema, the use of a compression bandage or garment during mild/moderate exercise was considered to be as important as using compression during resistance exercise. Although using compression bandaging or garments during the daytime was emphasized in the literature, it was estimated that 48.3% of patients suffering from upper-extremity lymphedema do not adhere to this recommendation [35]. Various reasons such as discomfort in the arm, functional difficulties in daily life activities, restriction of social participation, cosmetic reasons, having insufficient knowledge about the compression bandage benefits, financial conditions, and high pressure from multi-layered compression materials cause discontinuation of the use of compression bandages/garments [36]. Additionally, the obligation to wear compression during exercise has been identified as a barrier to regular exercise participation [37]. Al Onazi et al. [38] reported that wearing the compression bandage/garment for more than 12 h a day may not be necessary to control the severity of lymphedema, and there are many barriers to use. Another study showed that women with BCRL may experience practical and emotional problems with compression garments and have lower perception of HRQoL than those who do not wear the garment [3,39]. Therefore, lymphedema patients seek information about the acute effect on lymphedema if they continue their daily routine and exercise without compression for at least 24 h. In addition, patients who cannot receive long-term compression therapy or who are placed on long waiting lists for treatment request a short-term break from their treatment or receive short-term compression treatment [37]. In response to these problems experienced by patients, a few studies in the literature mention the acute effects of exercise along with compression during exercise after breast cancer treatments.

The International Society of Lymphology (ISL), the National Lymphedema Network (NLN), and most experts recommend the use of compressions, especially during vigorous physical activity, for BCRL in clinical settings [16,33,40,41]. There are studies in the literature reporting that participation in low-to-moderate-intensity aerobic and resistive exercise programs without compression sleeves do not acutely (immediately after exercise or at 24 h) precipitate or exacerbate lymphedema [34,42,43,44,45]. In another study [31], a transient increase in the severity of lymphedema according to the measurement of limb volume and extracellular water ratio was observed immediately after moderate-to-high-intensity resistance exercise sessions without compression sleeves, and there was a tendency towards reduced lymphedema 24 h post exercise. Bloomquist et al. [44] reported that low- and heavy-load resistance exercise without compression sleeves revealed similar acute responses in arm swelling according to BIS and dual-energy X-ray absorptiometry (DEXA) and related lymphedema symptoms in women with BCRL. While there was an increase in the L-Dex score and lymphedema symptoms immediately after a heavy-load session in 22% of the patients, increases had dissipated by the 24 h and 72 h follow-up. None of the patients with BCRL showed clinically significant increases in L-Dex 24 and 72 h after low-load resistance exercise without compression sleeves. In contrast to this report, we chose remedial exercises that are safer and lower in intensity in our study. All participants performed remedial exercises with and without compression bandaging, and it was found that acute changes in the extracellular water ratio and the relative arm volume and feelings of swelling, heaviness, and tightness remained relatively stable 24 h post exercise without compression.

Several studies investigated the acute effects of various types of exercise on BCRL. McNeely et al. [34] showed in a pilot study that the acute response to moderate-intensity endurance exercise in terms of changes in upper-extremity circumference measurement in women with breast cancer was similar, regardless of wearing compression sleeves during exercise, compared to before exercise. In our study, not only circumference measurement but also electrical impedance of extracellular fluid related to severity of lymphedema were evaluated. Additionally, to eliminate the effect of the 3.3% volume difference between the dominant and non-dominant arm volumes, correction for arm dominance was made and relative arm volume (%) was also evaluated in the analysis accordingly. The randomized controlled study of Lane et al. [46] revealed that after 2.5 min of 12 repeated sets of arm cranking exercise without compression sleeves, the lymphatic function assessed through lymphoscintigraphy had similar lymphatic clearance in patients with lymphedema and healthy controls. Similarly, the findings of the present study are consistent with evidence from the literature suggesting that participation in a 24 h remedial exercise program does not acutely increase the extent of swelling and lymphedema symptoms based on patient reports. However, we determined these similar findings with circumferential measurements and bioimpedance spectroscopy, which are objective measurement methods sensitive to changes in arm volume and extracellular water ratio. As a contribution to the existing literature, in the present study, we investigated remedial exercise as part of treatment for lymphedema. This study also demonstrated that while the symptoms of swelling, heaviness, and tightness remained relatively unchanged 24 h post remedial exercise without compression, it was found that all outcome measures showed significant improvement at 24 h following remedial exercise with compression bandaging. It could be considered that the combination of compression and remedial exercise have an extra effect on lymph flow from the muscle pump, acting as a frictional counterforce to the muscle contractions, thereby leading to what is presumed to have created and stimulated greater lymphatic flow [26].

Exercise-induced fatigue increases the extent of edema as it causes an increase in fluid overload in the lympho-venous system. It was reported that while breast cancer survivors knew they should exercise, exercise-induced fatigue and lymphedema are noted as barriers for taking regular exercise [47]. Even though the conservative advice for women with BCRL is to avoid exercise without a compression bandage/sleeve to prevent exacerbation, the remedial exercises performed without eliciting fatigue in this study did not acutely increase the extent of swelling or the severity of lymphedema symptoms. Concerning the possible mechanisms of the results of the present study, since the movements in remedial exercises have similar functionality during daily activities of the affected arm, it could be speculated that remedial exercises without compression do not cause overload to the lymphatic system, increasing the severity of lymphedema. According to the evidence obtained from the present study, women with BCRL are able to engage safely in remedial exercise without any acute increase in the severity of lymphedema within 24 h.

The strengths of this study include the fact that validated objective measurement methods sensitive to changes in the volume of the arm and extracellular water ratio were used. Additionally, the objective assessment of study outcomes using standardized procedures reduced the risk of measurement bias, and the ability to detect sensitive changes in symptoms was improved. In our study, the effects of remedial exercises in two different conditions on the same people were evaluated using the wash-out period. Therefore, we think that the responses are more reliable. Nevertheless, the present study also has some limitations. First, this study examined the 24 h effects of remedial exercise with and without compression on BCRL, but long-term intervention effects also need to be evaluated. Second, using the VAS for the measurement of self-reported symptoms of BCRL rather than a valid and reliable symptom scale specific to BCRL patients may be another limitation of our study. Evaluation of these symptoms will be helpful with quantitative and objective imaging methods. Hopefully, in the future, instruments capable of quantitatively measuring these factors will be developed. Although the sample size in our study was calculated based on the primary outcome measure of a previous study, the number of participants in this study was relatively small. The sample size can be calculated by considering secondary outcome measures in future studies on this subject. Finally, the present study focused on a population of breast cancer survivors. In terms of generalizability of the results, the acute effects of compression and remedial exercises should be investigated in different disease populations.

## 5. Conclusions

In conclusion, remedial exercises without compression do not increase lymphedema severity in the acute period. In addition, remedial exercises combined with compression bandaging produce a significant improvement in the acute phase. Based on these study results, remedial exercises can be performed by discontinuing compression for short 24 h periods without concern for worsening lymphedema. Additionally, remedial exercises accompanied by compression can be an acute therapy solution in patients who cannot receive long-term treatment or who will be placed on the waiting list. Further studies are required to assess the long-term effects of remedial exercises in the absence of compression.

## Figures and Tables

**Figure 1 healthcare-11-02949-f001:**
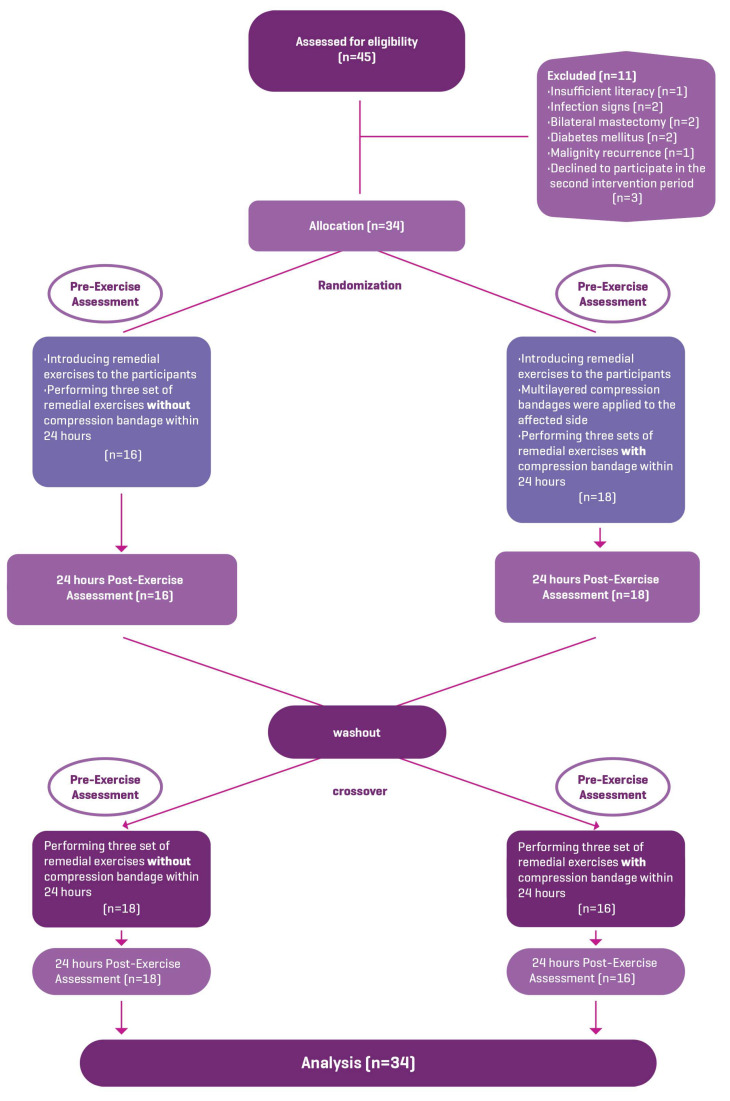
Study flow-chart.

**Table 1 healthcare-11-02949-t001:** Descriptive characteristics of the participants.

	N = 34 ^a^
Age (years)	59.8 ± 8.78
BMI (kg/m^2^)	29.6 ± 4.21
Education level (years)	9.8 ± 4.9
Type of surgery	
Breast-conserving surgery (lumpectomy) + ALND	5
Mastectomy + ALND	29
Dissected lymph nodes (n)	21.6 ± 6.8
Affected side	
Dominant (yes), n (%)	16 (47.05)
Non-dominant (yes) n (%)	19 (55.88)
Radiotherapy (yes %)	29 (91.1%)
Radiotherapy duration (n)	25 (21–26)
Chemotherapy (yes %)	31 (82.3%)
Number of cycles (n)	8 (6–12)
Time since surgery (months)	63.4 ± 39.3
Duration of lymphedema (months)	47.5 ± 34.9
Severity of lymphedema (moderate/severe)	15/19

^a^ Data are presented as mean ± standard deviation or number (%) or median (interquartile range). Abbreviations = BMI: body mass index, ALND: axillar lymph node dissection.

**Table 2 healthcare-11-02949-t002:** Pre- to 24 h post-exercise changes in limb volume and extracellular water ratio in patients with breast-cancer-related lymphedema.

Variables	Without Compression		*p*	With Compression		*p*
	Pre-Exercise	24 h Post-Exercise		Pre-Exercise	24 h Post-Exercise	
Affected total arm volume (cm^3^)	2868.58 ± 471.29	2880.35 ± 472.41	0.134	2872.27 ± 473.79	2805.44 ± 465.33	<0.001 *
Difference between affected and unaffected side (cm^3^)	547.85 ± 179.83	553.79 ± 197.30	0.506	548.08 ± 183.33	471.11 ± 173.05	<0.001 *
BIS (L-Dex score)	35.13 ± 14.08	35.79 ± 14.02	0.581	35.24 ± 14.38	32.07 ± 13.16	<0.001 *
Relative arm volume (%)	24.29 ± 8.85	24.56 ± 9.74	0.535	24.25 ± 8.93	20.80 ± 8.39	<0.001 *

Data are presented as mean ± standard deviation or number (%) or median (interquartile range). Abbreviations = BIS: bioimpedance spectroscopy. * Statistical significance level was set at *p* < 0.05.

**Table 3 healthcare-11-02949-t003:** Pre- to 24 h post-exercise changes in self-reported lymphedema symptoms in patients with breast-cancer-related lymphedema.

Self-Reported Symptoms Using VAS	Without Compression		*p*	With Compression		*p*
	Pre-Exercise	24 h Post-Exercise		Pre-Exercise	24 h Post-Exercise	
Swelling	6.11 ± 1.59	5.94 ± 1.72	0.361	6.24 ± 1.72	4.46 ± 1.45	<0.001 *
Heaviness	5.86 ± 2.20	6.24 ± 1.83	0.172	6.03 ± 1.91	4.25 ± 1.69	<0.001 *
Tightness	7.18 ± 1.61	6.81 ± 1.55	0.107	6.52 ± 1.48	4.86 ± 2.10	<0.001 *

Data are presented as mean ± standard deviation or number (%) or median (interquartile range). Abbreviations = VAS: visual analog scale. * Statistical significance level was set at *p* < 0.05.

**Table 4 healthcare-11-02949-t004:** Comparison results of percentage changes of remedial exercises with and without compression.

Variables	Amount of Change (%)		*p*
	Without Compression	With Compression	
Affected total arm volume (cm^3^)	0.1 (−0.6/1.4)	−2.4 (−2.9/−1.0)	<0.001 *
Difference between affected and unaffected side (cm^3^)	−1.6 (−5.7/5.0)	−13.9 (−20.7/−9.1)	<0.001 *
BIS (L-Dex score)	2.3 (−1.9/5.1)	−7.8 (−12.4/−3.8)	<0.001 *
Relative arm volume (%)	−1.5 (−5.7/5.2)	−13.8 (−20.8/−9.6)	<0.001 *
Swelling (VAS, cm)	−3.6 (−11.6/5.4)	−24.8 (−35.8/−15.1)	<0.001 *
Heaviness (VAS, cm)	1.4 (−15.1/45.2)	−24.0 (−43.7/−16.4)	<0.001 *
Tightness (VAS, cm)	−4.4 (−10.7/5.2)	−22.4 (−52.3/−2.1)	<0.003 *

Data are presented median (first quartile–third quartile). Percent change = [(Post-exercise value − Pre-exercise value)/Pre-exercise value] × 100. Abbreviations = BIS: bioimpedance spectroscopy, VAS: visual analog scale. * Statistical significance level was set at *p* < 0.05.

## Data Availability

The data supporting the findings of this study are available from the corresponding author upon reasonable request.

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
