# Peer review of "Acute Effects of Remedial Exercises with and without Compression on Breast-Cancer-Related Lymphedema"

_healthcare, 2023, doi:10.3390/healthcare11222949_

Round 1
Reviewer 1 Report
Comments and Suggestions for Authors
Dear Authors,
While the study addresses an important topic related to breast cancer-related lymphedema (BCRL) and the impact of remedial exercises, there are several aspects of the manuscript that raise concerns and warrant further clarification.
Major reviews:
INTRODUCTION: in my opinion, the introduction exhaustively describes the role of remedial exercise in the context of BCRL. However, the section could benefit from a more thorough exploration of the clinical implications of BCRL. Indeed, this is a complex condition with a pathogenesis which has not been fully understood yet. As modern advancements in breast cancer therapeutical strategies are currently improving survival rates, growing effort has been performed in current literature to address the disability burden of this condition. Accordingly, preventive strategies might contribute to improving not only the patient’s quality of life, but also economical resources allocation. You might consider implementing the section in light of the following suggestions:
- Muñoz-Alcaraz MN, Jiménez-Vílchez AJ, Pérula-de Torres LÁ, Serrano-Merino J, García-Bustillo Á, Pardo-Hernández R, González-Bernal JJ, González-Santos J. Effect of Conservative Rehabilitation Interventions on Health-Related Quality of Life in Women with Upper Limb Lymphedema Secondary to Breast Cancer: A Systematic Review. Healthcare (Basel). 2023 Sep 17;11(18):2568. doi: 10.3390/healthcare11182568.
- Nascimben M, Lippi L, de Sire A, Invernizzi M, Rimondini L. Algorithm-Based Risk Identification in Patients with Breast Cancer-Related Lymphedema: A Cross-Sectional Study. Cancers (Basel). 2023 Jan 4;15(2):336. doi: 10.3390/cancers15020336.
METHODS: as you stated, the randomization was performed by a computer program. While this is a suitable method for ensuring random allocation, it would be helpful to provide more specific details about the computer program used (e.g., name or type, version, and any relevant settings). Additionally, information on who operated the computer program and any measures taken to ensure allocation concealment (to prevent selection bias) should be included. Detailed reporting of the randomization process enhances transparency and the study's internal validity.
DISCUSSION: as you mentioned, the study focused on a population of breast cancer survivors who were recruited on a voluntary basis. This element might introduce a risk of self-selection bias, which should be discussed as a potential limitation of the present manuscript. Moreover, the limitations of the study should be addressed more exhaustively. For example, there is no discussion of the relatively small sample size or the limited generalizability due to the crossover design of this study.
Minor reviews:
ABSTRACT: please, rephrase the abstract in accordance with J’s instructions for authors. I.e., the abstract should be a total of about 200 words maximum. Moreover, it should be a single paragraph and should follow the style of structured abstracts, but without headings (i.e., background, methods, results, conclusions).
FIGURE 3: please, provide a higher resolution version of the flow diagram.
Author Response
Response 1.
Dear reviewer, thank you very much for your valuable comments. You are indeed right. These articles you mentioned are very beneficial and appropriate to the topic of our manuscript. According to your suggestions, we expanded the introduction based on these two articles. In addition, based on the information in Munoz et al.'s study, the introduction section has been revised by giving more examples of functional upper extremity disorders (such as sensory disorders, hypersensitivity) and psychological symptoms (such as poorer social well-being) caused by lymphedema. Because this systematic review also addresses the controversy about the impact of compression garment use in BCRL, we have also benefited from this systematic review in the discussion section of our manuscript. You can find all changes in the page and line numbers listed below.
Page 1. Paragraph 1. Line 32-37. Lymphedema is a progressive and persistent chronic inflammatory condition caused by damage or overload to the lymphatic system that negatively affects patients' work, career, and health-related quality of life (HRQoL), daily activities, and functionality [2,3]. Although lymphedema can occur in the breast, upper and lower arm, chest, and/or trunk of the affected side due to the mechanical insufficient lymphatic transport capacity, it most commonly appears in the upper extremity of the affected side [3].
Page 1-2. Paragraph 1. Line 45-51. According to a previous study which integrates a machine learning technology in determining risk factors in breast cancer survivors, number of metastatic lymph nodes, breast cancer grading, presence of lymphovascular invasion, type of estrogen therapy before breast cancer, age of the patient at diagnosis, trastuzumab therapy, human epidermal growth factor receptor 2 (HER2), and taxane-based chemotherapy have been characterized as the most important risk factors for BCRL development [8].
Page 2. Paragraph 1. Line 52. ….. sensory disturbances, hypersensitivity…
Page 2. Paragraph 1. Line 55….. poorer social well-being…
Page 9-10. Paragraph 2. Line 339-344. Al Onazi et al. [38] reported that wearing the compression bandage/garment for more than 12 hours a day may not be necessary to control severity of lymphedema and there are many barriers to use. Another study showed that women with BCRL may experience practical and emotional problems with compression garments and have lower perception of HRQoL than those who do not wear the garment [3,39].
Response 2.
Dear reviewer, thank you very much for your valuable comments. The information of the online computer program we used for block randomization is “Sealed Envelope Ltd, 27-31 Clerkenwell Close, London, UK”. Randomization was performed by Ceren Gursen, who had no role in data collection. Based on your comment, we revised the randomization in more detail. You can find all changes in the page and line numbers listed below.
Page 3. Paragraph 2. Line 119- 124. After baseline assessment, each participant completed the remedial exercise with a compression bandage and without compression bandaging in a randomized order separated by a 3-day wash-out period. Participants were randomized to determine which exercise intervention was initiated first in an allocation ratio of 1:1 using a computer-based block randomization procedure (Sealed Envelope Ltd, 27-31 Clerkenwell Close, London, UK) by C.G., who had no role in data collection.
Response 3.
Dear reviewer, thank you for pointing this out. We agree with this comment. You are very right about the potential limitations of the study you mentioned. Therefore, we added both limitations along with their reasons. These changes can be found;
Page 11. Paragraph 3. Line 427- 433. Although the sample size in our study was calculated based on the primary outcome measure of a previous study, the number of participants in this study was relatively small. The sample size can be calculated by considering secondary outcome measures in future studies on this subject. Finally, the present study focused on a population of breast cancer survivors. In terms of generalizability of the results, the acute effects of compression and remedial exercises should be investigated in different disease populations.
Response 4.
Dear reviewer, thank you very much for your valuable comments. You are right that the word count of the abstract section exceeded 200. Because we tried to summarize the purpose, method, results and conclusion presented in the main text, we tried to shorten the number of words as much as possible. The abstract has been rephrased a single paragraph with no headings.
Page 1. Line 12-27. Remedial exercises are an important part of the treatment for lymphedema, but there is little evidence to support the acute effects of remedial exercises with or without compression. The aim of this study was to investigate whether and how daily (performed within 24 hours) remedial exercises with and without compression bandaging acutely affect the severity of lymphedema and its symptoms in breast-cancer-related lymphedema (BCRL). In total, 34 patients with BCRL completed three sets of remedial exercises (within 24 hours) with and without compression bandaging in a randomized order separated by a 3-day wash-out period. The severity of lymphedema and extracellular water ratio were assessed before and 24 hours post exercise by using bilateral circumferential measurements and bioimpedance spectroscopy (in L-dex score), respectively, and the severity of self-reported symptoms (swelling, heaviness, and tightness) was assessed using a visual analogue scale. While there was no difference in all outcomes at 24 hours post exercise without compression (p >0.05), all outcomes decreased significantly compared to baseline at 24 hours after the exercise with compression (p<0.001). The remedial exercises performed in the absence of compression within 24 hours do not acutely increase the lymphedema and symptoms in BCRL. These are important preliminary findings, which can be used to inform future prospective evaluation of the long-term effects of remedial exercise performed without compression.
Response 5.
Dear reviewer, thank you for pointing this out. The study flow-chart was revised in terms of higher resolution. Additionally, according to your and other reviewer’s valuable suggestions, the chart was revised in more detail. We would like to also state that reviewer 3 suggested that we delete Figure 1 and 3 from the manuscript, because they provided no new information. Therefore, the flow-chart title was changes as Figure 1. This change can be found on Page 6.
5. Additional clarifications: Dear reviewer, first of all thank you very much for your all helpful comments and suggestions. We would like to state that the manuscript was edited in terms of extensive English editing service by MDPI after your and other reviewer’s valuable revisions was completed.

Reviewer 2 Report
Comments and Suggestions for Authors
In introduction. “Exercise interventions are considered the mainstay for the treatment of lymphedema, and remedial exercises form an integral component in the complex decongestive physiotherapy (CDP), which is the gold standard approach in the treatment of lymphedema”
Surgical treatments (reconstructive approach) should be acknowledged in this section. The following recent article is dealing with combined surgical and conservative treatment of lymphedema. It should be mentioned as a recent approach to BCRL.
Ciudad P, Bolletta A, Kaciulyte J, Losco L, Manrique OJ, Cigna E, Mayer HF, Escandón JM. The breast cancer-related lymphedema multidisciplinary approach: Algorithm for conservative and multimodal surgical treatment. Microsurgery. 2023 Jul;43(5):427-436. doi: 10.1002/micr.30990.
Materials and methods
The type of study should be clearly stated. Is this a prospective study?
161-162. Paragraph “interventions”: “The two interventions comprising remedial exercises without compression bandage and Remedial exercises with or without compression bandage were performed by all par-ticipants in a randomized order separated by a 3-day wash-out period”.
Here two cohort without compression bandage were reported. Please revise
Figure 3: the two treatment groups (16 and 18 women are not displayed). Two boxes report 34 patients instead. Please revise
Discussion line 294-320
This part is a bit tortuous. A revision of syntax should be accomplished, this paragraph should be also a bit shortened as it is a way too long.
Comments on the Quality of English LanguageSyntax and grammar revision should be accomplished.
These are some examples:
- “A total of 34 patients with 16 BCRL were completed 3 sets of remedial exercises (within 24 hours) with and without compression 17 bandage in a randomized order separated by a 3-day wash-out period.”
- “Since it is more practical remedial exercises to patients compared to mentioned exercise approaches, they are more widely used in lymphedema management”.
- Line 235-238 should be revised
- Similarly, in the findings of this present study are consistent with evidence
And elsewhere in the manuscript. A thorough revision should be performed to improve the ease of reading
Author Response
Response 1.
Dear reviewer, thank you very much for your valuable comments. You are indeed right. The management of BCRL requires combined conservative and surgical treatment approach to be multidisciplinary. According to your valuable suggestion, we have mentioned current surgical approaches to BCRL based on this article. Additionally, the paragraph following the paragraph mentioned about surgical approaches has also been revised in order to provide the intended meaning.
You can find the changes in the page and line numbers listed below:
Page 2. Paragraph 2. Line 58-65. Since lymphedema is a complex condition, multidisciplinary approaches combining conservative and surgical techniques are required in the treatment of BCRL. In the surgical management of BCRL, physiological techniques (lymphatic-venous anastomosis and vascularized lymph node transfer) are more effective in mild/moderate lymphedema, while ablative techniques are more necessary in severe lymphedema characterized by trophic skin changes such as fibrosis, thickness, and warty overgrowths [11]. In order to preserve postoperative results, it is recommended to combine pre- and postoperative conservative modalities with both physiological and ablative surgical techniques [11,12].
Page 2. Paragraph 3. Line 66-73. Currently, several conservative interventions exist for lymphedema management in the literature; however, there is limited consensus on the optimal conservative modality of BCRL [12]. Among the conservative approaches, complex decongestive physiotherapy (CDP), consisting of patient education, manual lymphatic drainage, compression therapy, remedial exercises, and skin care, is the most recommended method to improve HRQoL [3,11]. Additionally, exercise interventions such as supervised aerobic exercise and progressive resistance exercise have been reported to have significant potential effects in the prevention of BCRL [13].
Response 2:
Dear reviewer, thank you very much for pointing this out. Yes, this study was designed as a prospective, randomized and crossover design. We have stated design of the study in Materials and Methods section.
Page 3. Paragraph 1. Line 100. This study utilized a prospective, randomized, and cross-over design
Response 3.
Thank you for your valuable correction. First of all, we apologize for the mistake of the sentence. We made the necessary correction. You can find the changes in the page and line numbers listed below:
Page 4. Paragraph 3. Line 170-172. The two interventions comprising remedial exercises without compression bandaging and remedial exercises with compression bandaging were performed by all participants in a randomized order separated by a 3-day wash-out period.
Response 4.
Dear reviewer, thank you very much for your valuable comment. First of all, we apologize for the misunderstanding. We would like to explain that the randomization was performed to determine which exercise intervention participants would perform first. According to randomization, sixteen people first completed the remedial exercise program without compression and then the remedial exercise program with compression, while eighteen people first completed the remedial exercise program with compression and then the remedial exercise program without compression bandage. To make this clear in the working diagram, we revised the study flow-chart for your comment.
We would like to also state that reviewer 3 suggested that we delete Figure 1 and 3 from the manuscript, because they provided no new information. Therefore, the flow-chart title was changes as Figure 1.
Page 5. Paragraph 4. Line 229-231. Sixteen participants were randomized to begin remedial exercises without a compression bandage first, and eighteen were randomized to begin with a compression bandage first.
Page 6. Figure 1. Study flow-chart
Response 5.
Thank you for your valuable correction. First of all, we apologize for the duplication in the paragraph you mentioned. Accordingly your valuable comment, we have shortened and revised this paragraph.
Page 9. Paragraph 2. Line 324- 329. Until the early 2000s, the use of compression bandages or garments during waking hours and physical activities was the essential issue in the management of BCRL [33]. Additionally, women with or at risk of BCRL were advised to avoid strenuous upper-extremity activity and/or resistance exercise, especially including the affected upper extremity because of the risk that such activity could lead to the exacerbation of lymphedema [34].
Additional clarifications: Dear reviewer, first of all thank you very much for your all helpful comments. We would like to state that the manuscript was edited in terms of extensive English editing service by MDPI after we had completed your valuable revisions/corrections.

Reviewer 3 Report
Comments and Suggestions for Authors
The paper presents the results of a randomized cross-over trial to assess immediate effect of remedial exercises in postmastectomy arm lymphedema patients. Effect was assessed if multi-layered compression bandage was used and when there was no compression.
The study is well-designed and well-conducted. Conclusions are supported by findings. I see no serious objections. Only some minor remarks have to be addressed.
1. I recommend to shorten Introduction by shortening of its last paragraph. The text from line 63 to about lines 74-75 is more suitable for Discussion section.
2. Figure 1 have to be deleted. It brings no useful information as the images like this can be easily found using search engines.
3. The same is for Figure 3. How multi-layered bandage looks like is well known and can be easily found everywhere on the internet. This image brings no new information.
4. Flow-chart has to be restructured. After Allocation box there should be two boxes for compression and no compression. Then there have to be a common box (ore separate boxes) for evaluation. And then – another two boxes showing that intervention was crossed over.
5. The title of the flow chart is better to change for Study flow-chart
6. Table 1. No need to mention that N is for number. It’s common for every paper.
7. Tables 2, 3 and 4. As you mention the p-level in the Materials and methods there is no need to repeat under each table.
8. Table 3. Please, mention that numbers in boxes are obtained by VAS scale. The same is for Table 4 regarding swelling, heaviness and tightness.
Comments on the Quality of English Language-
Author Response
Response 1.
Dear reviewer, thank you very much for your valuable comments. We agree with this comment. Therefore, the lines you mentioned in the introduction section have been shortened and moved to the discussion section. The relevant paragraph in the introduction and discussion has been revised to clarify the meaning. You can find the changes in the page and line numbers listed below:
Page 2. Paragraph 4. Line 83-85. The clinical guidelines recommend that exercise should be performed with compression in order not to increase lymphedema; however, the evidence behind this clinical recommendation is unclear [16].
Page 10. Paragraph 2. Line 352-361. The International Society of Lymphology (ISL), the National Lymphedema Network (NLN), and most experts recommend the use of compressions, especially during vigorous physical activity, for BCRL in clinical settings [16,33,40,41]. There are studies in the literature reporting that participation in low-to-moderate-intensity aerobic and resistive exercise programs without compression sleeve do not acutely (immediately after exercise or at 24 hours) precipitate or exacerbate lymphedema [34,42-45]. In another study [31], a transient increase in the severity of lymphedema according to the measurement of limb volume and extracellular water ratio was observed immediately after moderate-to-high-intensity resistance exercise sessions without compression sleeve and there was a tendency towards reduced lymphedema 24 hours post exercise.
Response 2.
Dear reviewer, thank you very much for your valuable comment. We agree with this comment. We have deleted Figure 1 and its reference in the sentence.
You can see the change on Page 4. Line 159, indicating that we deleted the Figure 1.
Response 3.
Dear reviewer, thank you very much for your valuable suggestion. You are indeed right. Figure 3 and the reference of the figure in text were deleted because it brings no new information, as you stated.
You can see the changes on Page 5. Line 202, indicating that we deleted the Figure 3.
Response 4.
Dear reviewer, thank you very much for your valuable comment. You are indeed right. We revised the flow-chart to show the study design more clearly, as you mentioned.
You can found the revised study flow-chart on Page 6.
Response 5.
Dear reviewer, thank you for your feedback. As per your valuable advice, we changed the title of the flowchart to study flow-chart. These changes can be found on the page number and line indicated below:
Page 5. Paragraph 4. Line 229. Figure 1 shows the study flow-chart.
Page 6. Line 259. Figure 1. Study flow-chart
Response 6.
Dear reviewer, thank you very much for your feedback. According to your suggestion, we deleted the mention of “N=number” in the abbreviations. This change can be found on the page number and line indicated below:
Page 7. Line 262. Table 1. Abbreviations = BMI: body mass index, ALND: axillar lymph node dissection.
Response 7.
Dear reviewer thank you for pointing this out. You are right that we repeated the p level under each table. We have deleted the statement of “Statistical significance level was set at p < 0.05” accordingly your valuable suggestion. These changes can be found on the page number indicated below:
Page 8. (Table 2 and 3)
Page 9. (Table 4)
Response 8.
Dear reviewer, thank you very much for your valuable correction. We revised the variables in table 3 to indicate that they were obtained from VAS. In addition, we have also added (VAS, cm) next to the swelling, heaviness and tightness variables in the Table 4, as you mentioned. These changes can be found on the page number and line indicated below:
Page 8. (Table 3) We highlighted the changes as “Self-reported symptoms using VAS”
Page 9. (Table 4) “Swelling (VAS, cm), Heaviness (VAS, cm), Tightness (VAS, cm)”
Additional clarifications: Dear reviewer, first of all thank you very much for your all helpful comments. The manuscript was edited in terms of extensive English editing service by MDPI after we had completed your valuable revisions/corrections.

Round 2
Reviewer 1 Report
Comments and Suggestions for Authors
Dear Authors,
I have assessed the latest form of the manuscript. In my opinion, the overall quality of the paper has significantly improved after newly added insightful comments and improvements.
In my opinion, the paper would be suitable for publication in its present form, as I have no further suggestions.
Best regards
Reviewer 2 Report
Comments and Suggestions for Authors
The authors revised the manuscript according to the comments